# Single-cell monitoring of dry mass and dry mass density reveals exocytosis of cellular dry contents in mitosis

**Teemu P Miettinen**[1,2]*, **Kevin S Ly**[3], **Alice Lam**[3], **Scott R Manalis**[1,2,3,4]*

[1]Koch Institute for Integrative Cancer Research, Massachusetts Institute of Technology, Cambridge, United States; [2]MIT Center for Precision Cancer Medicine, Massachusetts Institute of Technology, Cambridge, United States; [3]Department of Biological Engineering, Massachusetts Institute of Technology, Cambridge, United States; [4]Department of Mechanical Engineering, Massachusetts Institute of Technology, Cambridge, United States

**Abstract** Cell mass and composition change with cell cycle progression. Our previous work characterized buoyant mass dynamics in mitosis (Miettinen et al., 2019), but how dry mass and cell composition change in mitosis has remained unclear. To better understand mitotic cell growth and compositional changes, we develop a single-cell approach for monitoring dry mass and the density of that dry mass every ~75 s with 1.3% and 0.3% measurement precision, respectively. We find that suspension grown mammalian cells lose dry mass and increase dry mass density following mitotic entry. These changes display large, non-genetic cell-to-cell variability, and the changes are reversed at metaphase-anaphase transition, after which dry mass continues accumulating. The change in dry mass density causes buoyant and dry mass to differ specifically in early mitosis, thus reconciling existing literature on mitotic cell growth. Mechanistically, cells in early mitosis increase lysosomal exocytosis, and inhibition of lysosomal exocytosis decreases the dry mass loss and dry mass density increase in mitosis. Overall, our work provides a new approach for monitoring single-cell dry mass and dry mass density, and reveals that mitosis is coupled to extensive exocytosis-mediated secretion of cellular contents.

*For correspondence:
teemu@mit.edu (TPM);
srm@mit.edu (SRM)

## Editor's evaluation

The authors measure dry mass, dry volume and the density of the dry mass in growing and proliferating mammalian cells at high temporal resolution and with high precision. Using this method to study mitotic cells, the authors show that some cells lose dry mass early in mitosis by a mechanism involving exocytosis. This work improves upon the authors' method to measure the mass of single cells and its thought-provoking conclusion is that dividing cells 'clean out' their contents to give the daughter cells a fresh start.

## Introduction

Cells coordinate the activity of biosynthetic and catabolic pathways as well as the uptake and secretion of components in order to maintain appropriate molecular composition. In continuously proliferating cells, these processes are coordinated so that all cellular components are doubled during each cell cycle, whereas differentiating cells may change their composition to match their functions (*Cadart et al., 2019*; *Ho et al., 2018*; *Lloyd, 2013*; *Miettinen et al., 2017*; *Schmoller and Skotheim, 2015*). Changes in cells' molecular composition are typically studied with approaches such as mass

spectrometry, which can quantify the molecular details of a cell population after lysis. However, non-invasive methods, which can monitor the cellular composition of live single cells with high resolution, are better suited for the study of dynamic and temporary events such as mitosis.

Cells' total molecular content, that is, dry mass, can be monitored using methods such as quantitative phase microscopy (QPM). Dry mass measurements are critical for understanding cell size and growth regulation, but they are often limited in resolution and, more importantly, they do not inform us about the composition of the dry mass. One approach that overcomes this is stimulated Raman scattering microscopy, which can measure the total amount of proteins and lipids in a live cell (*Figueroa et al., 2021*; *Oh et al., 2020*). While Raman scattering-based methods provide spatial resolution and details about the molecular changes taking place, they can also cause phototoxicity, which significantly limits the temporal resolution when tracking the same cell(s) over time (*Zhang et al., 2021*). Alternatively, QPM-based dry mass measurements can be coupled to microscopy-based total volume measurements to quantify dry density (dry mass/total volume) (*Cooper et al., 2013*; *Odermatt et al., 2021*; *Yeo et al., 2021*; *Zlotek-Zlotkiewicz et al., 2015*). These methods include cell's water content in the total volume quantification. For example, a decrease in the amount of cellular lipids decreases the dry density of a cell, as the cell has less dry mass per total volume.

Here, we introduce a new approach for monitoring single cell's dry mass (i.e. total mass − water mass), dry volume (i.e. total volume − water volume), and density of the dry mass (i.e. dry mass/dry volume), which we will refer to as dry mass density. Cell's dry mass, dry volume, and dry mass density can be quantified by comparing the cell's buoyant mass in normal water ($H_2O$) and heavy water (deuterium oxide, $D_2O$)-based solutions (*Feijó Delgado et al., 2013*). However, high concentration of $D_2O$ prevents cell proliferation and can result in cell death (*Kampmeyer et al., 2020*; *Schroeter et al., 1992*; *Takeda et al., 1998*). Here, we show that we can expose the cell to $D_2O$ only periodically (every 75 s) and for very short durations (10–15 s), thus allowing us to non-invasively monitor the dry mass and dry mass density of the same cell over many generations without compromising cell growth and viability. By definition, the measured dry mass density is independently of the cell's water content and therefore sensitive toward changes in the relative molecular composition (dry composition), not the overall concentration of dry mass. For example, a decrease in the amount of cellular lipids increases the dry mass density of a cell, as lipids are lower in dry mass density than other cellular components.

We have previously monitored the buoyant mass accumulation of single mammalian cells in mitosis (*Miettinen et al., 2019*) using the suspended microchannel resonator (SMR), a cantilever-based single-cell buoyant mass sensor (*Burg et al., 2007*). We found that buoyant mass accumulation rate is high from prophase to metaphase-anaphase transition, that is, early mitosis. In contrast, studies using QPM have suggested that dry mass does not accumulate in early mitosis (*Zlotek-Zlotkiewicz et al., 2015*), or that cells may even experience a decrease in dry mass in early mitosis (*Liu et al., 2020*; *Liu et al., 2022*). If cells actually lose dry mass in mitosis, this would indicate previously unrecognized compositional changes taking place in mitosis. Although the exact dynamics of dry mass accumulation in mitosis remain debatable, our study addresses this seeming contradiction between buoyant and dry mass measurements by providing a high-resolution view of the mitotic dry mass and dry mass density changes.

## Measurement method
### Measurement principle

In our approach, we consider the buoyant mass of a cell to be dependent on two distinct physical 'sections' of the cell, the dry content and the water content. To measure the cell's dry content independently of the water content, we measure the cell's buoyant mass in $H_2O$ and $D_2O$-based solutions. Under these conditions, the influence of the water content on buoyant mass can be excluded, because the intracellular water is exchanged with extracellular water, making the intracellular water content neutrally buoyant with extracellular solution. This allows us to detect the cell's dry mass (i.e. total mass − water mass), dry volume (i.e. total volume − water volume), and dry mass density (i.e. dry mass/dry volume). Importantly, our approach assumes that all water within the cell is exchangeable between $H_2O$ and $D_2O$. Accordingly, our dry volume measurement is distinct from the excluded cell volume detected by measuring cell volume following strong hyperosmotic shocks, which does not remove all water from the intracellular space.

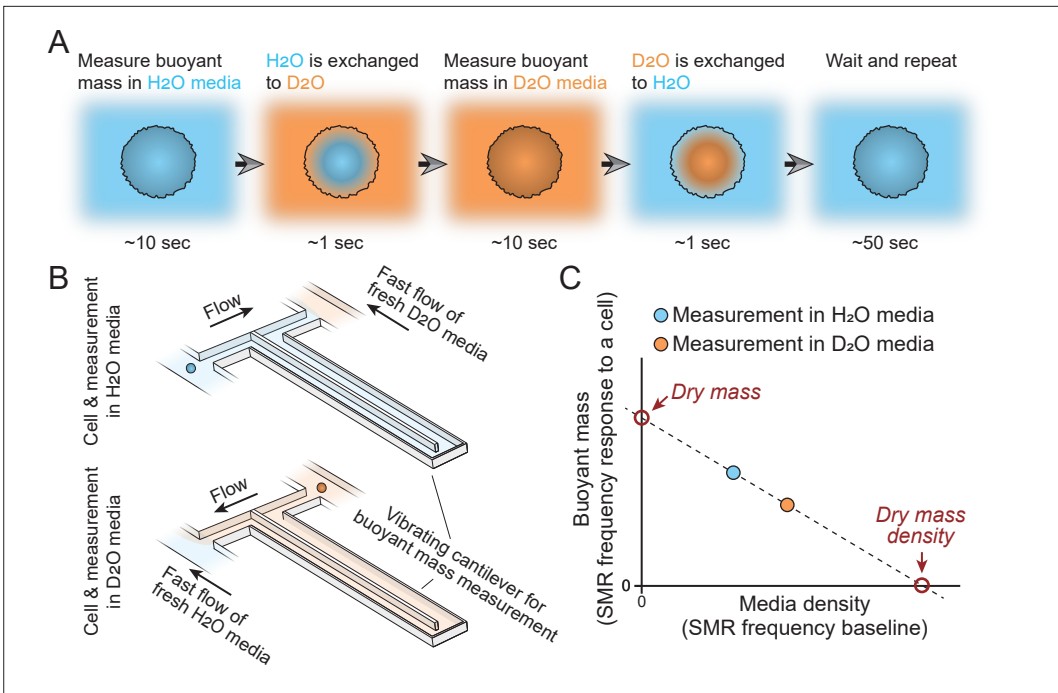

**Figure 1.** Schematic of dry mass and dry mass density measurements using the suspended microchannel resonator (SMR). (**A**) The cell's buoyant mass is first measured in normal, $H_2O$-based culture media (blue), after which the cell mixes with $D_2O$-based culture media (orange). The water content inside the cell exchanges to match the external water content. The cell's buoyant mass is then measured again in the $D_2O$-based media and the cell is mixed with normal, $H_2O$-based media, where the cell waits for the next measurement. (**B**) In practice, these measurements are carried out using an SMR, where $H_2O$ and $D_2O$-based medias are kept on different sides of the cantilever. Continuous flushing of fresh media into the system prevents the two fluids from equilibrating over time. The cell is depicted as a small blue/orange sphere. (**C**) To calculate the dry mass and dry mass density of the cell, the two buoyant mass measurements are correlated as a function of media density. The SMR's baseline signal (vibration frequency) is proportional to media density and this measurement is obtained immediately before and after each buoyant mass measurement (frequency response to a cell flowing through the SMR) to account for variability in media mixing.

The online version of this article includes the following figure supplement(s) for figure 1:

**Figure supplement 1.** Quantifications of dry mass and dry mass density measurement errors.

**Figure supplement 2.** Dry mass and dry mass density measurements are not limited by the rate of water exchange.

In practice, the measured cell first travels through the SMR while filled with $H_2O$-based solution to obtain a buoyant mass measurement (*Figure 1A*). On the other side of the SMR, the cell is briefly exposed to $D_2O$-based solution. Cells constantly exchange their water content, causing intracellular $H_2O$ to be exchanged to $D_2O$. The cell then travels back through the SMR in $D_2O$ solution to obtain a second buoyant mass measurement (*Figure 1B*). The densities of the extracellular fluid are not constant due to fluid mixing between measurements, but this is accounted for by directly measuring the extracellular solution density, that is, the baseline frequency of the SMR, together with every buoyant mass measurement. These buoyant mass measurements can then be used to calculate the dry mass, dry volume, and dry mass density of the cell (*Figure 1C*; *Feijó Delgado et al., 2013*).

Mathematically, the model for our measurement can be written out as:

$$m_b = m_{dry}\left(1 - \frac{\rho_{fluid}}{\rho_{dry\ mass}}\right) + V_{water}\left(\rho_{water} - \rho_{fluid}\right) \tag{1}$$

where $m_b$ is the buoyant mass of the cell in a given fluid, $m_{dry}$ is the dry mass of the cell, $\rho_{fluid}$ is the density of the fluid outside the cell, $\rho_{dry\ mass}$ is the dry mass density, $V_{water}$ is the volume of intracellular water content, and $\rho_{water}$ is the density of intracellular water content. The first term of *Equation 1* describes the contribution of the cell's dry mass to buoyant mass and the latter term describes

the contribution of the cell's water content to buoyant mass. As $H_2O$ and $H_2O$-based media are very similar in density (1.00 g/ml), as are $D_2O$ and $D_2O$-based media (1.10 g/ml), the density of water inside the cell is near identical to the density of fluid outside the cell whether the measurement is carried out in $H_2O$ or $D_2O$-based solution (see *Feijó Delgado et al., 2013*, for error estimations). Consequently, the latter term of *Equation 1* approaches zero for both the $H_2O$ and the $D_2O$-based buoyant mass measurements and the two consecutive measurements can be used to solve the first term of the equation, that is, the dry mass and dry mass density of a cell.

## Measurement precision and stability

We characterized our measurement precision and stability using several approaches. First, we monitored live non-growing L1210 cells (mouse lymphocytic leukemia cells) by cooling our measurement system down to +4°C to prevent growth and compositional changes. We measured the same cell repeatedly and defined our measurement error as the coefficient of variation during a 1 hr trace. With this approach our measurement errors in dry mass and dry mass density were ~1.3% and ~0.3%, respectively (*Figure 1—figure supplement 1A&B*). We did not observe measurement drifting even during a longer experiment (*Figure 1—figure supplement 1A*), suggesting a high level of measurement stability. Second, we monitored fixed cells at +37°C and observed measurement errors that were close to those seen in live cells at +4°C (*Figure 1—figure supplement 1C*). Third, we monitored the dry mass and density of 10 µm diameter polystyrene beads at +37°C. This suggested similar dry mass measurement errors as the previous experiments with live and fixed cells, but lower errors in dry mass density measurements (*Figure 1—figure supplement 1D*). We therefore estimate our dry mass measurement error to be ~1.3%, which is comparable to current state-of-the-art QPM measurements (*Liu et al., 2020*; *Reed et al., 2011*). For dry mass density measurements, we estimate our measurement error to be ~0.3%. We are not aware of any other single-cell methods capable of quantifying this metric. Furthermore, as we carry out dry mass and dry mass density measurements approximately every 75 s, we can further increase our precision by averaging multiple measurements while still capturing changes on timescales meaningful for studying dynamic and transient events such as mitosis.

## Measurement sensitivity to intracellular water exchange

Our measurement principle involves the cells exchanging their intracellular water content between $H_2O$ and $D_2O$, and this exchange has to be complete for our measurements to reflect the dry mass and dry mass density of cells. Direct experimental evidence suggests that cells normally exchange their water content within ~1 s (*Potma et al., 2001*; *Quirk et al., 2003*; *Zhao et al., 2008*). To support this, we carried out back-of-the-envelope calculations on the intracellular water exchange times, as based on reported aquaporin water permeability (*Gravelle et al., 2013*; *Hashido et al., 2007*; *Yang and Verkman, 1997*), abundance (*Denker et al., 1988*), and a cell water volume of ~1000 fl (approximate water volume for an L1210 cell). This suggested that cells are capable of exchanging all of their water volume within 0.1–2.5 s, depending on the reported aquaporin water permeability. Notably, the true intracellular water exchange time may differ from this due to intracellular water diffusion, water molecule interactions with other biomolecules, and additional means of water transport in and out of the cell (additional aquaporin isoforms, endo- and exocytosis, etc.).

In our typical measurement, a cell is exposed to $D_2O$ for 10–15 s, which should enable complete water exchange. To validate the completeness of the water exchange, we correlated the measured dry mass and dry mass density to the time that a cell was exposed to the $D_2O$-based media between each buoyant mass measurement. Whether examining end-point measurements of single L1210 cells in a population (*Figure 1—figure supplement 2A*), a single-cell trace of a live L1210 cell cooled to +4°C (*Figure 1—figure supplement 2B*) or a single-cell trace of an L1210 cell at +37°C (*Figure 1—figure supplement 2C*), the measured dry mass and dry mass density were independent of the time in $D_2O$-based media. Thus, the rate of intracellular water exchange is not limiting our measurements, as also suggested by previous results in bacteria and yeast (*Feijó Delgado et al., 2013*).

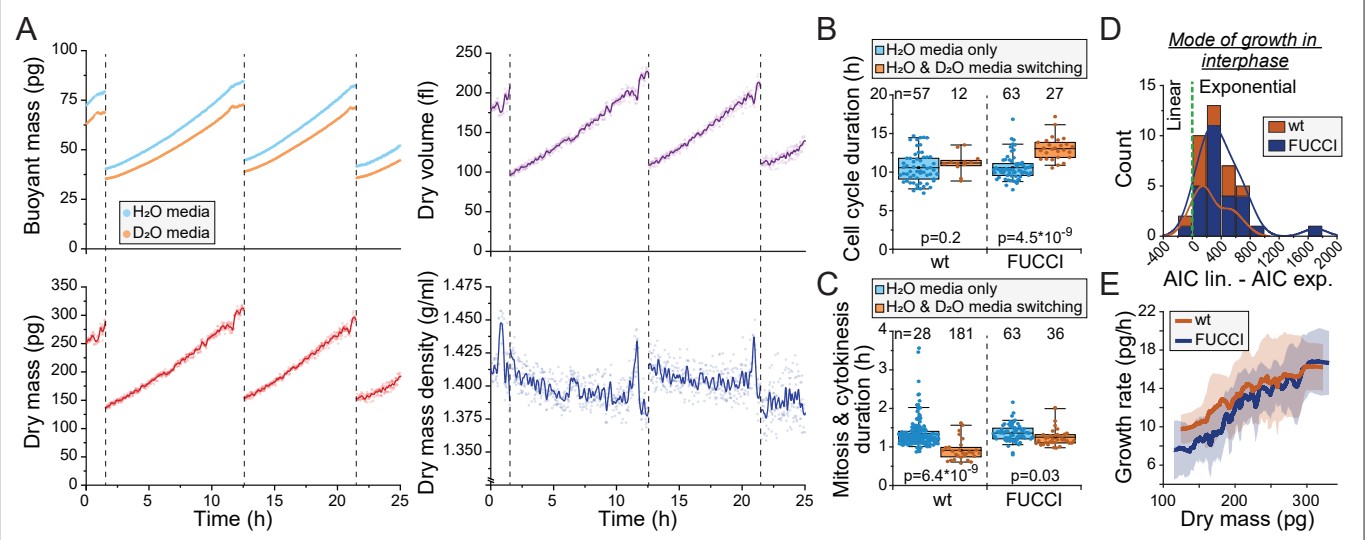

**Figure 2.** Monitoring buoyant mass, dry mass, dry volume, and dry mass density of single cells. (**A**) An example dataset of an ancestral wild-type (wt) L1210 lineage tracked for its dry mass, dry volume, and dry mass density over two full cell cycles. Cell divisions are indicated with dashed vertical lines. Opaque points represent individual measurements and the solid lines represent smoothened data. (**B**) Cell cycle durations of wt and FUCCI L1210 cells grown in the suspended microchannel resonator (SMR) in normal media (blue) or with periodic $D_2O$ exposure (orange). Boxplot line: mean; box: interquartile range; whiskers: 5–95% range. p-Values calculated using Welch's t-test; n values refer to individual cells. (**C**) Same as (**B**), but data displays the combined duration of mitosis and cytokinesis. (**D**) Histogram of Akaike information criterion (AIC) of a linear fit – AIC of an exponential fit for wt and FUCCI L1210 cells in interphase (n = 12 and 27 cells, respectively). Datapoints (individual cells) with positive values are better described by exponential rather than linear growth. (**E**) Cell dry mass growth rate as a function of dry mass in wt and FUCCI L1210 cells in interphase (n = 12 and 27 cells, respectively). The line and shaded area represent mean ± SD.

The online version of this article includes the following figure supplement(s) for figure 2:

**Figure supplement 1.** Influence of $D_2O$ on cell cycle durations in normal culture.

**Figure supplement 2.** Dry mass and dry mass density measurements are not causing systematic drifts in cell size at division.

**Figure supplement 3.** Dry and buoyant mass correlate near-perfectly, except in mitosis.

## Results

### Monitoring single cell's dry mass and dry mass density

In order to continuously monitor the dry mass and dry mass density of single cells, we needed to periodically expose the cells to $D_2O$, which can impact cell growth and viability. We first tested the sensitivity of L1210, BaF3 (mouse pro-B cell myeloma), and S-Hela (human adenocarcinoma) cells toward different concentrations of $D_2O$ by monitoring cell proliferation with live cell imaging. Under normal culture conditions, all three cell lines can tolerate continuous exposure to up to 30% $D_2O$ without changes to proliferation rate (*Figure 2—figure supplement 1*). Cells were still able to proliferate at higher $D_2O$ concentrations, albeit slowly.

With the observed level of $D_2O$ tolerance, we hypothesized that we could monitor the dry mass density of a single cell growing inside the SMR as long as we minimize $D_2O$ exposure. To achieve this, we maintain the cell most of the time in the $H_2O$-based media (*Figure 1A*), and our $D_2O$-based media contains only 50% $D_2O$. With this approach we were able to monitor the dry mass, dry volume, and dry mass density of L1210 cells throughout multiple cell cycles (*Figure 2A*). We then examined if our measurement approach influences normal cell growth. Under these conditions, cell cycle durations of wild-type (wt) L1210 cells remained comparable to what we have observed previously in the SMR in the absence of $D_2O$ (*Miettinen et al., 2019*; *Mu et al., 2020*; *Figure 2B*). However, when examining FUCCI (hGeminin-mKO) cell cycle reporter expressing L1210 cells, we observed slightly increased cell cycle durations. In addition, we observed that the duration of mitosis and cytokinesis was systematically shorter when cells were periodically exposed to $D_2O$ than what we have previously observed in other SMR experiments (*Figure 2C*).

Previous results examining mammalian cell growth in buoyant mass, dry mass, or volume have all showed that cell growth in interphase is, on average, exponential (*Cadart et al., 2018*; *Liu et al., 2020*; *Mu et al., 2020*). For dry mass growth, this has only been examined using QPM. Utilizing our new dry mass growth measurements, we examined the mode of growth in wt and FUCCI L1210 cells in interphase. We compared linear and exponential growth models using Akaike information criterion (AIC) and found that exponential fits are a better model for our data in 37 out of 39 interphases (*Figure 2D*). Consistent with exponential growth, instantaneous dry mass growth rates (pg/hr) increased as a function of dry mass (*Figure 2E*). We then examined the stability of cell size homeostasis in ancestral L1210 lineages with ≥3 full cell cycles. We did not observe systematic changes in cell's dry mass at division (*Figure 2—figure supplement 2A*). Together, these results show that we can monitor the dry mass and dry mass density of single cells over long time periods using our heavy water-based measurement approach, and while limited $D_2O$ exposure may influence cell proliferation rate, it does not radically alter cell size and growth regulation.

## Dry and buoyant mass correlate near-perfectly in interphase but not in mitosis

A common limitation of cell size and growth studies is the reliance on a single type of size measurement – for example, a cell's dry mass or total volume. This can result in seemingly conflicting findings if cells change their composition. Our data enabled direct comparison of dry mass and buoyant mass measurements. We first examined the correlation between dry mass and buoyant mass on population level by carrying out end-point measurements of single cells. We observed a high degree of correlation ($R^2$ >0.97) for L1210, BaF3, and S-Hela cell populations (*Figure 2—figure supplement 3A*). We then correlated dry mass and buoyant mass within our L1210 single-cell growth traces and observed a very high degree of correlation in interphase of each cell ($R^2$~0.99), but lower and more variable correlations in mitosis and cytokinesis ($R^2$~0.5–0.9) (*Figure 2—figure supplement 3B&C*). Together, these observations indicate that buoyant mass is an accurate proxy for cell's dry mass in interphase, but not necessarily in mitosis.

Cells' dry mass and buoyant mass will correlate if dry mass density does not change, assuming that measurements are carried out in normal cell culture media or other water-based solution with density close to water (see *Equation 1*). Indeed, in our population level data, dry mass density displayed little variability, with coefficient of variability around 1.6% for L1210, BaF3, and S-Hela cells (*Figure 2—figure supplement 3D*). This is approximately 15-fold less variability than observed for dry mass. The low variability in dry mass density within a proliferating cell population represents the tight regulation of cells' molecular composition.

## Cells transiently lose dry mass and increase dry mass density in early mitosis

We then examined dry mass and dry mass density behavior(s) in mitosis. We detected mitotic stages based on acoustic scattering from the cell, which reports approximate mitotic entry and the metaphase-anaphase transition (*Kang et al., 2019*; *Figure 3—figure supplement 1*). We also validated the metaphase-anaphase transition timing by examining the degradation of the hGeminin-mKO cell cycle reporter in the FUCCI L1210 cells (*Figure 3—figure supplement 1*). Curiously, we observed that many, but not all, L1210 cells lost dry mass and, especially, dry volume following mitotic entry, resulting in increased dry mass density in early mitosis (*Figure 3A–C*; *Figure 3—source data 1*). This was observed in both wt and FUCCI L1210 cells, although the FUCCI cells displayed larger changes in mitosis (*Figure 3B*). On average, the FUCCI L1210 cells lost ~4% of dry mass and increased dry mass density by ~2.5%, and these changes took place in approximately 15 min (*Figure 3C*). In the most extreme cases, cells lost ~8% of their dry mass while increasing dry mass density by ~4%. These changes were transient, as cells recovered their lost dry mass and decreased their dry mass density at metaphase-anaphase transition (*Figure 3B*). The recovery of dry mass at metaphase-anaphase transition was faster than the dry mass growth observed prior to mitosis, suggesting that this 'recovery growth' is distinct from normal cell growth. After the metaphase-anaphase transition, dry mass growth continued during cytokinesis. The total amount of dry mass accumulated in mitosis and cytokinesis was large, over 10% of the dry mass accumulated in the whole cell cycle (*Figure 3D*). The extent of mitotic dry mass loss in early mitosis was coupled to the extent of corresponding dry mass density

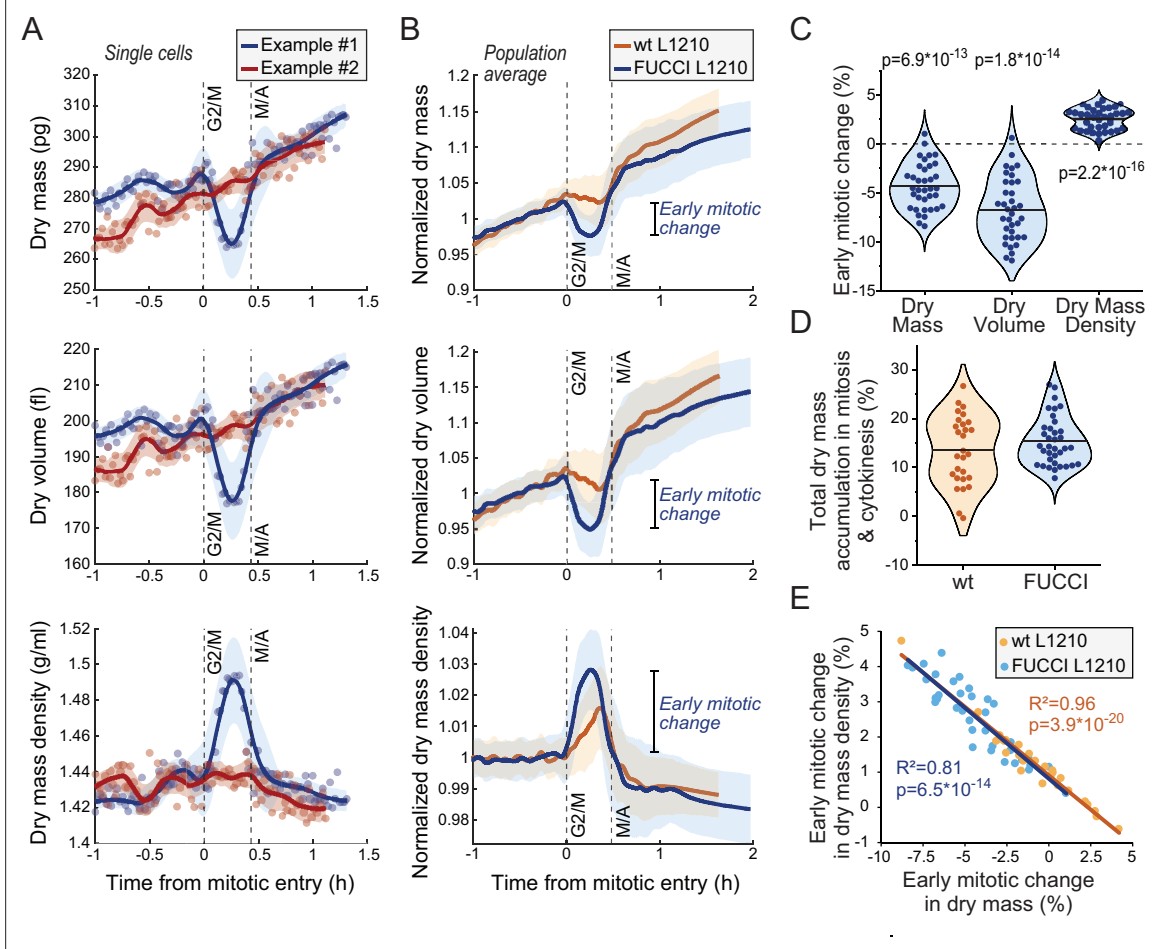

**Figure 3.** Cells transiently lose dry mass and increase dry mass density in early mitosis. (**A**) Two example FUCCI L1210 cells with different dry mass (top), dry volume (middle), and dry mass density (bottom) behaviors in early mitosis. Opaque points represent individual measurements; thick line and shaded area represent smoothened data (mean ± SD). Dashed vertical lines indicate approximate G2/M and metaphase-anaphase (M/A) transitions. (**B**) Population average dry mass, dry volume, and dry mass density behavior in wild-type (wt) and FUCCI L1210 cells. Thick line and shaded area represent mean ± SD. N = 31 cells from 19 independent experiments for the wt cells; N = 36 cells from 13 independent experiments for the FUCCI cells. (**C**) Quantifications of dry composition changes in early mitosis in FUCCI L1210 cells. Dots represent individual cells; horizontal line represents mean; data is same as in panel (**B**); p-values calculated using one sample t-test and represent difference from zero. (**D**) Dry mass accumulation from mitotic entry to cell division relative to the dry mass accumulated in the whole cell cycle. Dots represent individual cells; horizontal line represents mean; data is same as in panel (**B**). (**E**) Correlation between dry mass and dry mass density change in early mitosis for wt and FUCCI L1210 cells. Data is same as in panel (**B**); correlation p-values calculated using ANOVA. Raw data can be found in *Figure 3—source data 1*.

The online version of this article includes the following source data and figure supplement(s) for figure 3:

**Source data 1.** Dry mass and dry mass density traces for different cell lines and conditions.

**Figure supplement 1.** Cell cycle indicators and mitotic dry composition change.

**Figure supplement 2.** Cell-to-cell variability in the loss of dry mass in early mitosis reflects biological, non-genetic cell-to-cell variability.

**Figure supplement 3.** Mitotic dry mass and dry mass density behavior in BaF3 and THP-1 cells.

increase (*Figure 3E*). Importantly, the increase in dry mass density uncoupled buoyant mass and dry mass measurements (*Equation 1*), causing buoyant mass to increase in early mitosis even in the cells which lost dry mass (*Figure 3—figure supplement 2A-C*).

## Mitotic dry mass and dry mass density changes display large, non-genetic cell-to-cell variability

The early mitotic dry mass and dry mass density changes were not observed uniformly across all wt and FUCCI L1210 cells (*Figure 3C*). Approximately 7% of all L1210 cells measured (10 out of 67 cells)

displayed little to no dry mass density increase in early mitosis. We also observed mother-daughter pairs where the mother cell displayed mitotic changes in its dry mass and dry mass density but the daughter cell did not, or vice versa (*Figure 3—figure supplement 2A-C*). The difference in mitotic behavior between individual cells did not correlate with experiment-specific noise levels (*Figure 3—figure supplement 2D*). Together, these results indicate a large degree of non-genetic cell-to-cell variability in the regulation of mitotic cell mass and composition. Furthermore, since the dry mass and dry mass density changes were not present in all cells, they are not required for a successful cell division.

We also monitored the dry mass and dry mass density of BaF3 and THP-1 (human acute monocytic leukemia) cells through mitosis. The BaF3 cells displayed a large degree of cell-to-cell variability in mitotic behavior. While some individual BaF3 cells lost ~4% of their dry mass during early mitosis, on average the BaF3 cells did not lose dry mass in mitosis but growth rather stopped during early mitosis, which was followed by a rapid recovery at metaphase-anaphase transition (*Figure 3—figure supplement 3A-E*). In THP-1 cells, on average, cells lost ~5% of their dry mass, which was also recovered at metaphase-anaphase transition (*Figure 3—figure supplement 3F-J*). In both cell types, dry mass density transiently increased in early mitosis. Thus, the mitotic loss of dry mass and increase in dry mass density are not specific to L1210 cells. More broadly, our dry mass measurements suggest that early mitosis is coupled to previously unrecognized secretion of cellular components.

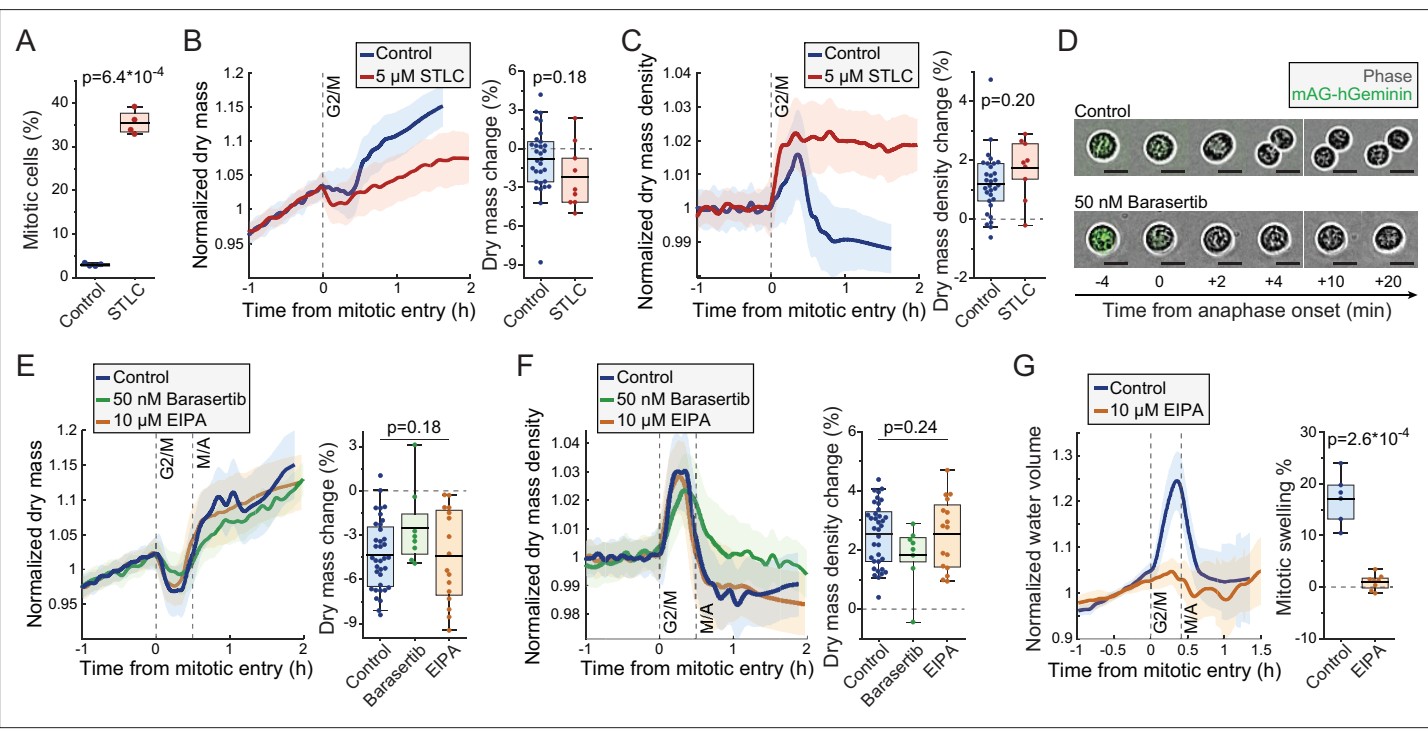

**Figure 4.** Mitotic dry mass loss and dry mass density increase do not require morphological changes. (**A**) % of mitotic cells in a wild-type (wt) L1210 population following 5 hr control or *S*-trityl-L-cysteine (STLC) treatment. $N = 4$ independent cultures; p-value calculated using Welch's t-test. (**B**) Normalized dry mass behavior for control and STLC-treated wt L1210 cells (left) and quantifications of early mitotic dry mass changes (right). $N = 31$ cells from 19 independent experiments for control; $N = 9$ cells from 9 independent experiments for STLC; p-value calculated using Welch's t-test. (**C**) Same as panel (**B**), but for dry mass density. (**D**) Representative phase contrast and mAG-Geminin reporter images of control and Barasertib-treated FUCCI L1210 cells in mitosis. $N > 20$ cells from three independent experiments. (**E**) Normalized dry mass behavior for FUCCI L1210 cells treated with indicated chemicals (left) and quantifications of early mitotic dry mass changes (right). $N = 36$ cells from 13 independent experiments for control; $N = 8$ cells from eight independent experiments for Barasertib; $N = 16$ cells from 15 independent experiments for EIPA; p-value calculated using ANOVA. (**F**) Same as panel (**E**), but for dry mass density. (**G**) Normalized intracellular water volume behavior for control and EIPA-treated FUCCI L1210 cells (left) and quantifications of mitotic cell swelling (right). $N = 6$ cells from five independent experiments for control; $N = 8$ cells from eight independent experiments for EIPA; p-value calculated using Welch's t-test. In dry mass, dry mass density, and water volume traces, the thick line and shaded area represent mean ± SD; boxplot line: mean; box: interquartile range; whiskers: 5–95% range. Raw data can be found in *Figure 3—source data 1*.

The online version of this article includes the following figure supplement(s) for figure 4:

**Figure supplement 1.** L1210 cell population dry mass density following mitotic perturbations.

## Mitotic dry mass and dry mass density changes do not require morphological changes

We investigated the mechanistic basis for the mitotic changes in dry mass and dry mass density using L1210 cells as our model system. We first examined the influence of mitotic progression and mitotic morphological changes to cell's dry mass. We arrested wt L1210 cells to prometaphase using 5 µM kinesin inhibitor *S*-trityl-L-cysteine (STLC) and validated mitotic arrest by quantifying the fraction of mitotic cells using DNA and p-histone H3 (Ser10) labeling (*Figure 4A*). When monitoring STLC-treated cells, we observed slightly larger, but statistically not significantly different, loss of dry mass and increase in dry mass density upon mitotic entry as in control cells (*Figure 4B&C*). However, the mitotically arrested cells did not recover their dry mass or dry mass density, and following the initial changes the STLC-treated cells displayed slow dry mass accumulation and steady dry mass density for the rest of the mitotic arrest. These results verify that the increase in dry mass density and decrease in dry mass take place before the completion of prometaphase, and that mitotic progression is required for the rapid recovery of dry mass and dry mass density.

As the recovery of dry mass and dry mass density required mitotic progression, we asked if the recovery is coupled to cell elongation and furrowing. To this end, we inhibited cytokinesis in FUCCI L1210 cells using 50 nM Aurora B inhibitor Barasertib. We validated the inhibition of cytokinesis by imaging cell morphology as cells proceeded through metaphase-anaphase transition, as indicated by the degradation of the hGeminin-mKO signal (*Figure 4D*). The Barasertib-treated cells displayed slightly lower, but statistically not significantly different, loss of dry mass and increase in dry mass density as untreated FUCCI L1210 cells (*Figure 4E&F*). The duration of mitosis was lengthened in some Barasertib-treated cells, resulting in slower recovery of dry mass and dry mass density. Thus, the mitotic changes in cell's dry mass and dry mass density cannot be fully explained by cell elongation and furrowing, although cell elongation and furrowing (or Aurora B activity) could still have an influence on dry mass and dry mass density dynamics in mitosis.

Another major morphological change that mitotic cells undergo is the mitotic cell swelling, where cells take up water and can increase their total volume by 10–20% (*Son et al., 2015*; *Zlotek-Zlotkiewicz et al., 2015*). This takes place simultaneously with the dry mass changes, that is, cells swell from prophase to metaphase and recover in anaphase. To investigate if mitotic cell swelling is required for the dry mass and dry mass density changes, we inhibited mitotic swelling using 10 µM sodium-hydrogen exchanger inhibitor 5-(*N*-ethyl-*N*-isopropyl)-amiloride (EIPA). We first validated that EIPA prevents mitotic cell swelling by monitoring approximate intracellular water volume (*Son et al., 2015*). This revealed that the FUCCI L1210 cells swell significantly in early mitosis, but this is near completely prevented by EIPA treatment (*Figure 4G*). However, when we monitored the dry mass and density of FUCCI L1210 cells in the presence of EIPA, we did not observe any changes in the amount of dry mass lost or dry mass density increased in early mitosis when compared to control cells (*Figure 4E and F*). Therefore, the mitotic dry mass and dry mass density changes do not require mitotic cell swelling. However, it remains possible that these two phenotypes share a common driving mechanism. Finally, we validated that the mitotic perturbations used here did not influence the typical dry mass density of the cells (*Figure 4—figure supplement 1*).

## Lysosomal exocytosis is increased in early mitosis

Cells undergo a constant influx and efflux of dry mass through the flux of ions, metabolites, proteins, and other components that are taken up and secreted by a cell. When evaluating potential mechanisms for the mitotic loss of dry mass, we first considered the extent of dry mass loss in mitosis, which was ~8% in extreme cases. Macromolecules would be the most likely form of mass lost, as they constitute most of cell's dry mass. In contrast, inorganic ions are an unlikely form of dry mass lost, as these constitute only ~1% of cell's dry mass (*Cooper and Hausman, 2013*). Cells could secrete metabolites or small molecules in mitosis, but at least the secretion of lactate, a key metabolite secreted by cells, is not increased in mitotic L1210 cells (*Kang et al., 2020*). Furthermore, it seems unlikely that cells could lose large amounts of ions and/or small molecules while increasing osmotic pressure, which is required for the mitotic cell swelling. We therefore focused our study on mechanisms that secrete macromolecules.

Macromolecule secretion is largely due to exocytosis, the process where intracellular vesicles or organelles merge with the plasma membrane to release their contents to the extracellular space.

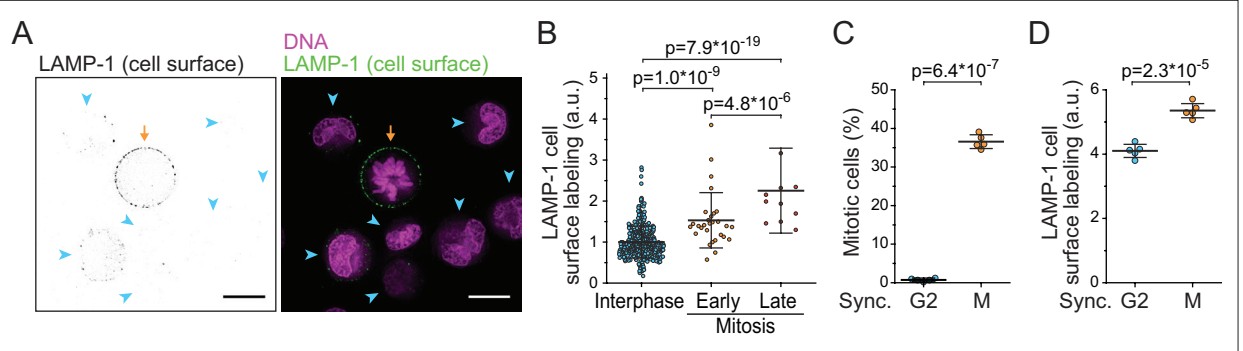

**Figure 5.** Lysosomal exocytosis is increased in early mitosis. (**A**) A representative image of surface LAMP-1 immunolabeling alone (left) and together with DNA labeling (right) in live L1210 cells. Orange arrow indicates a mitotic cell, blue arrowheads indicate interphase cells. Scale bars depict 10 μm. (**B**) Microscopy quantifications of cell surface LAMP-1 immunolabeling in unsynchronized L1210 cells. Early mitosis refers to prophase, prometaphase, and metaphase; late mitosis refers to anaphase. N = 380, 29, and 11 cells for interphase, early, and late mitotic cells, respectively; data pooled from two independent experiments; p-values calculated using ANOVA followed by Sidakholm post hoc test. (**C**) % of mitotic cells in L1210 cell populations following synchronization to G2 or early mitosis. N = 5 independent cultures; p-value calculated using Welch's t-test. (**D**) Flow cytometry quantifications of L1210 cell population surface LAMP-1 immunolabeling following synchronization to G2 or early mitosis. N = 5 independent cultures; p-value calculated using Welch's t-test. In all figures, line and whiskers indicate mean ± SD.

The online version of this article includes the following figure supplement(s) for figure 5:

**Figure supplement 1.** Additional examples of live L1210 cell surface LAMP-1 labeling.

**Figure supplement 2.** Lysosomal exocytosis is higher in early mitosis than in G2 in BaF3 and THP-1 cells.

This is balanced by the reverse process called endocytosis. Most studies have found that endocytosis stops in mitosis, more specifically from mitotic entry until anaphase (*Fielding et al., 2012*; *Schweitzer et al., 2005*; *Warren et al., 1984*). This corresponds to the timing of dry mass loss that we have observed. However, contradictory results on mitotic endocytosis have also been reported (*Boucrot and Kirchhausen, 2007*). Changes in exocytosis have not been studied in similar detail during mitosis, but lysosomal exocytosis, where cells secrete lysosomal contents by fusing lysosomes with the plasma membrane (*Reddy et al., 2001*), has been suggested to increase in late mitosis (*Nugues et al., 2018*). Notably, lysosomal exocytosis has also been shown to secrete fatty acids (which are low in dry mass density) out of the cell (*Cui et al., 2021*), which could increase the dry mass density of a cell.

We hypothesized that the loss of dry mass in early mitosis is driven by increased exocytosis of components such as lysosomes. Lysosomal exocytosis, and the fusion of lysosomes with the plasma membrane, can be quantified by monitoring the presence of the lysosomal membrane protein LAMP-1 on cell surface (*Reddy et al., 2001*; *Samie and Xu, 2014*). We immunolabeled LAMP-1 in freely proliferating, live L1210 cells and validated the surface-specific labeling using microscopy (*Figure 5A*; *Figure 5—figure supplement 1*). Quantifications of the microscopy indicated that mitotic cells had more LAMP-1 on their surface than interphase cells (*Figure 5B*). Furthermore, the amount of LAMP-1 on cell surface increased from early mitosis (prophase, prometaphase, and metaphase) to late mitosis (anaphase) (*Figure 5B*), indicating that cell surface LAMP-1 accumulates throughout early mitosis. This is in contrast to the previous report suggesting that lysosomal exocytosis increases specifically in late mitosis (*Nugues et al., 2018*).

To further support the observation that lysosomal exocytosis is increased in early mitosis, we synchronized L1210 cells to G2 and early mitosis using RO-3306 (CDK1 inhibitor) and STLC, respectively (*Figure 5C*). Notably, STLC arrests cells in prometaphase, where the mitotic dry mass changes are taking place. We then immunolabeled cell surface LAMP-1 and quantified the labeling using flow cytometry. Cells arrested to early mitosis displayed higher levels of LAMP-1 on their surface than cells arrested to G2 (*Figure 5D*). We also examined cell surface LAMP-1 immunolabeling in BaF3 and THP-1 cells after cell cycle synchronizations to G2 or early mitosis. In both cell lines, LAMP-1 labeling was higher in early mitosis than in G2 (*Figure 5—figure supplement 2*). Thus, lysosome exocytosis is increased in early mitosis.

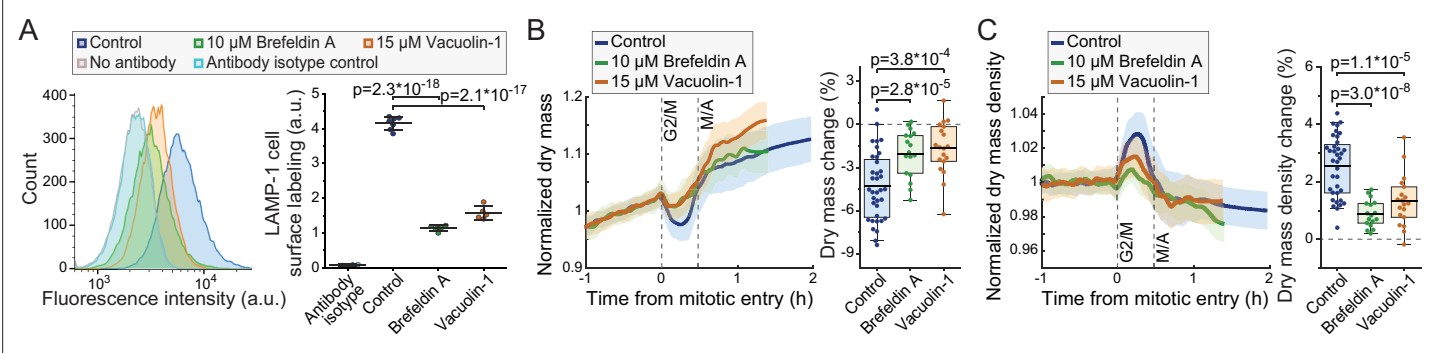

**Figure 6.** Inhibitors of lysosomal exocytosis decrease mitotic dry mass secretion. (**A**) Representative histograms (left) and quantifications (right) of live L1210 cell surface LAMP-1 immunolabeling following indicated, 4 hr long chemical treatments. Line and whiskers indicate mean ± SD; N = 4–7 independent cultures. (**B**) Normalized dry mass behavior for FUCCI L1210 cells treated with indicated chemicals (left) and quantifications of early mitotic dry mass changes (right). N = 36 cells from 13 independent experiments for control; N = 17 cells from 17 independent experiments for brefeldin A; N = 19 cells from 19 independent experiments for vacuolin-1; p-values calculated using ANOVA followed by Sidakholm post hoc test; Thick line and shaded area represent mean ± SD; boxplot line: mean; box: interquartile range; whiskers: 5–95% range. (**C**) Same as panel (**B**), but for dry mass density. Raw data can be found in *Figure 3—source data 1*.

The online version of this article includes the following figure supplement(s) for figure 6:

**Figure supplement 1.** L1210 cell population dry mass density following exocytosis perturbations.

## Inhibitors of lysosomal exocytosis decrease mitotic dry mass loss and dry mass density increase

We then investigated if inhibition of lysosomal exocytosis prevents the dry mass and dry mass density changes taking place in mitosis. For this, we utilized 10 µM brefeldin A, an inhibitor of vesicle-mediated protein trafficking and exocytosis (*Hendricks et al., 1992*), and 15 µM vacuolin-1, a small molecule inhibitor of lysosomal exocytosis (*Cerny et al., 2004*). The ability of vacuolin-1 to preventing lysosomal exocytosis has been controversial (*Cerny et al., 2004*; *Cui et al., 2021*; *Huynh and Andrews, 2005*), and we first validated that lysosomal exocytosis is inhibited by brefeldin A and vacuolin-1. Both chemicals prevented the accumulation of LAMP-1 on plasma membrane in L1210 cells (*Figure 6A*; *Figure 5—figure supplement 1*), indicating that lysosomal exocytosis was inhibited.

We then monitored the dry mass and dry mass density of FUCCI L1210 cells in the presence of brefeldin A or vacuolin-1. Both chemicals decreased the mitotic changes to cells' dry mass and dry mass density (*Figure 6B&C*) without preventing cell divisions. The chemicals did not cause clear changes to the baseline dry mass density of cells (*Figure 6—figure supplement 1*). Thus, lysosomal exocytosis is, at least partly, responsible for the loss of dry mass and increased dry mass density in early mitosis. However, other mechanisms, including other forms of exocytosis, may also be involved, because inhibition of lysosomal exocytosis did not prevent all dry mass loss in mitosis, and because our results are limited to a single cell line.

## Discussion

Here, we have developed a method using the SMR to monitor the dry mass, dry volume, and dry mass density of suspension grown cells over long periods of time with high precision. To the best of our knowledge, this is the first method capable of non-invasively measuring cell's dry volume or dry mass density. We anticipate several uses for this method. First, our dry mass density measurements provide an orthogonal approach for studying the macromolecular composition of cells. While dry mass density measurements lack molecular specificity, they are non-invasive, quantitative, and have high temporal resolution. Together, these qualities make our measurements valuable for studying the dynamics of molecular composition changes, especially in situations where rapid changes cannot be easily resolved by more typical population level measurements, such as mass spectrometry. Second, our method provides an additional approach for measuring dry mass. QPM has several advantages over our method, including higher throughput and suitability for adherent cells, but our method is comparable in precision and has high temporal resolution. Our method is also not limited by phototoxicity.

This enables us to observe dry mass dynamics that may be difficult to capture with other methods. Third, the ability of our method to measure cellular dry volume could enable precise, specific, and non-invasive quantifications of single-cell water content, provided it is coupled with a measurement of total cell volume.

Using our method, we have expanded our previous study of mammalian cell growth dynamics in mitosis. Importantly, the mitotic dry mass dynamics differ from our previously reported buoyant mass dynamics due to a transient change in dry mass density in early mitosis (*Equation 1*). This finding reconciles our previous mitotic buoyant mass measurements (*Miettinen et al., 2019*) with the QPM-based dry mass measurements by others (*Liu et al., 2020*; *Zlotek-Zlotkiewicz et al., 2015*). The overall dry mass growth remains high in mitosis and cytokinesis, and for L1210 cells over 10% of the total dry mass growth in the cell cycle takes place in mitosis and cytokinesis. This supports our previous main conclusion that average cell growth rates in mitosis and cytokinesis are comparable to those observed in interphase, which is consistent with the high protein synthesis rates reported for unperturbed mitosis (*Coldwell et al., 2013*; *Miettinen et al., 2019*; *Stonyte et al., 2018*; *Sun et al., 2019*). In interphase, dry mass density is conserved, making buoyant mass an excellent proxy for dry mass.

A key discovery made here is the significant loss of dry mass in early mitosis. Importantly, a loss of dry mass in mitosis has been reported in adherent mammalian cells studied using QPM (*Liu et al., 2020*; *Liu et al., 2022*), but the higher temporal resolution of our results enables us to examine this phenotype in more detail. However, our measurement approach decreased the duration of mitosis and cytokinesis, which could potentially influence the dry mass dynamics we observe in mitosis. This decrease in mitotic duration is contradictory to previous studies which show that mitosis is prolonged by $D_2O$ exposure (*Lamprecht et al., 1990*; *Takahashi and Sato, 1983*), possibly due to the brief but repeated $D_2O$ exposure in our measurements. Our results reveal a high degree of cell-to-cell variability in the mitotic loss of dry mass and cells can divide without losing mass. Mechanistically, we show that exocytosis of lysosomes is increased in early mitosis and the loss of dry mass is, at least partly, dependent on lysosomal exocytosis. Other mechanisms, especially other forms of exocytosis, may also be involved, as we were not able to completely block the dry mass loss in early mitosis by preventing lysosomal exocytosis. In addition, the loss of dry mass in early mitosis could be partly explained by decreased endocytosis, as the total dry mass of a cell is defined by the balance between uptake and secretion processes, such as endo- and exocytosis, respectively. Our results also show that cells are able to recover dry mass at a high rate at metaphase-anaphase transition. Interestingly, the exocytosis of lysosomes is typically followed by endocytosis, which removes lysosomal membranes from plasma membrane (*Idone et al., 2008*; *Tam et al., 2010*). Such a compensatory endocytosis could explain the rapid recovery of dry mass at metaphase-anaphase transition, but further quantitative studies on endo- and exocytosis are needed to confirm this.

The magnitude of dry mass density increase in mitosis was large. We have previously observed similar magnitude changes in dry mass density when perturbing proliferation in mammalian cell (*Feijó Delgado et al., 2013*). To provide some rough estimates of what kind of compositional changes would be required to achieve the dry mass loss and dry mass density increase, we carried out a back-of-the-envelope calculations. Assuming a typical mammalian cell composition and typical macromolecule dry mass densities (*Alberts, 2008*; *Feijó Delgado et al., 2013*), we calculated the degree of lipid loss needed to increase dry mass density by 2.5%. This suggested that cells would have to secrete ~1/3 of their lipid content in early mitosis. This could be achieved via lysosomal exocytosis of lipids. Lipid droplets, the main lipid storages inside cells, are frequently trafficked into and degraded in lysosomes (*Singh et al., 2009*), and lipid droplets can also be secreted via lysosomal exocytosis (*Minami et al., 2022*). However, it seems likely that the mitotic dry mass density increase also involves secretion of other low dry mass density components (e.g. lipoproteins, specific metabolites) and/or a minor, transient increase in high dry mass density components (e.g. RNAs, specific proteins) in early mitosis. Indeed, CDK1 activity has been suggested to drive a transient increase in protein and RNA content in early mitosis (*Asfaha et al., 2022*; *Clemm von Hohenberg et al., 2022*; *Miettinen et al., 2019*; *Shuda et al., 2015*).

Why would cells exocytose lysosomal contents and a significant amount of total cellular biomass in mitosis? We hypothesize three potential reasons for this: (1) Lysosomal exocytosis acts in plasma membrane repair (*Reddy et al., 2001*), and lysosomal exocytosis could enable a temporarily increase

in plasma membrane area for mitotic cell swelling. (2) Lysosomal exocytosis is used for secretion of various proteins and enzymes into the extracellular space (*Blott and Griffiths, 2002*), and mitotic cells may secrete extracellular matrix modifying enzymes in order to ensure an appropriate physical milieu for cell division. (3) Lysosomal exocytosis has been shown to clear the cell of excessive/toxic cellular proteins and lipids, including α-synuclein (*Tsunemi et al., 2019*) and tau (*Xu et al., 2021*). Mitotic lysosomal exocytosis may act as a 'reset' that enables the daughter cells to be born with a minimal load of useless/harmful contents. Consequently, non-dividing cells, such as neurons, could be more prone to suffer from excessive/toxic protein and lipid buildup, and the mechanism(s) that promote(s) mitotic lysosomal exocytosis could act as candidate drug target(s) for promoting cellular clearance of toxic/excess components. Lysosomal exocytosis is also perturbed in a variety of lysosomal and lipid storage disorders (reviewed in *Samie and Xu, 2014*; *Settembre and Ballabio, 2014*), and lysosomal exocytosis can drive the cellular expulsion of chemotherapeutics out of cancer cells (*Vyas et al., 2022*; *Zhitomirsky and Assaraf, 2017*). Understanding the mechanism(s) of mitotic lysosomal exocytosis could have implications for these diseases as well.

# Materials and methods

## Key resources table

| Reagent type (species) or resource | Designation | Source or reference | Identifiers | Additional information |
|---|---|---|---|---|
| Cell line (*Mus musculus*) | wt L1210 | ATCC | Cat#CCL-219 | |
| Cell line (*Mus musculus*) | FUCCI L1210 | Other | | Generated in a previous study (*Son et al., 2015*, *Nature Methods*), cells originate from ATCC (Cat#CCL-219) |
| Cell line (*Mus musculus*) | BaF3 | RIKEN BioResource Center | Cat#RCB4476 | |
| Cell line (*Homo sapiens*) | S-HeLa | Other | | Kindly provided by laboratory of Kevin Elias from Brigham And Women's Hospital |
| Cell line (*Homo sapiens*) | THP-1 | Other | | Kindly provided by laboratory of Jianzhu Chen from Massachusetts Institute of Technology |
| Cell line (*Mus musculus*) | wt L1210 | ATCC | Cat#CCL-219 | |
| Chemical compound, drug | STLC | Sigma-Aldrich | Cat#164739; CAS:2799-07-7 | |
| Chemical compound, drug | Barasertib | Cayman Chemical | Cat#13600; CAS:639089-54-6 | Alternative Names: MK 0457, VX 680 |
| Chemical compound, drug | EIPA | Cayman Chemical | Cat#14406; CAS:1154-25-2 | |
| Chemical compound, drug | RO-3306 | Cayman Chemical | Cat#15149; CAS:872573-93-8 | All experiments were done using a stock under 2 weeks of age |
| Chemical compound, drug | Vacuolin-1 | Cayman Chemical | Cat#20425; CAS:351986-85-1 | |
| Chemical compound, drug | Brefeldin A | Cayman Chemical | Cat#11861; CAS:20350-15-6 | |
| Antibody | Anti-LAMP-1 conjugated to Alexa Fluor 488 (rat monoclonal) | Thermo Fisher Scientific | Cat#53-1071-82; RRID:AB_657536 | Clone: eBio1D4B (1D4B) (1:50 dilution) |
| Antibody | IgG2a kappa Isotype Control conjugated to Alexa Fluor 488 (rat monoclonal) | Thermo Fisher Scientific | Cat#53-4321-80; RRID:AB_493963 | Clone: eBR2a (1:50 dilution) |
| Antibody | Anti-LAMP-1 conjugated to Alexa Fluor 488 (mouse monoclonal) | Thermo Fisher Scientific | Cat#53-1079-42; RRID:AB_2016657 | Clone: eBioH4A3 (1:50 dilution) |

*Continued on next page*

*Continued*

| Reagent type (species) or resource | Designation | Source or reference | Identifiers | Additional information |
|---|---|---|---|---|
| Other | RPMI media, powder | Thermo Fisher Scientific | Cat#31800022 | Cell culture media |
| Other | Heavy water (D$_2$O) | Sigma-Aldrich | Cat#151882; CAS:7789-20-0 | Heavy water used for culture media |
| Other | NUCLEAR-ID Red DNA stain | Enzo Life Sciences | Cat#ENZ-52406 | DNA stain (1:250 dilution) |
| Other | 10 µm Diameter Duke Standards 2000 Series Uniform Polystyrene Particles | Thermo Fisher Scientific | Cat#2010A | Standard particles for calibration |
| Software, algorithm | MATLAB R2017b and R2021b | MathWorks | | Used to analyze the SMR raw data and generate data plots |
| Software, algorithm | Origin 2021b | OriginLab | | Used to perform statistical analyses and generate data plots |

## Cell culture and media composition

L1210, BaF3, THP-1, and S-Hela cells were cultured in RPMI media (Thermo Fisher Scientific, #11835030) supplemented with 10% FBS, 1 mM sodium pyruvate, 10 mM HEPES, and antibiotic/antimycotic. This served as the H$_2$O-based media for experiments and the D$_2$O-based media was made to an identical composition using RPMI media powder (Thermo Fisher Scientific, #31800022) and heavy water (Sigma-Aldrich, #151882). D$_2$O-based media pH was adjusted to 7.4 with HCl and NaOH, and the media was filtered with 0.2 µm filter. To minimize cell exposure to heavy water in the SMR experiments, the D$_2$O-based media was diluted with the normal H$_2$O-based media to contain 50% heavy water. All experiments were started using cells growing exponential at a confluency of 250.000–500.000 cells/ml.

All cell lines were tested negative for mycoplasma. L1210 cells were obtained from and authenticated by ATCC. BaF3 cells were obtained from and authenticated by RIKEN BioResource Center. THP-1 cells were kindly gifted by the laboratory of Jianzhu Chen from Massachusetts Institute of Technology, and S-Hela cells were kindly gifted by the laboratory of Kevin Elias from Brigham and Women's Hospital. No additional authentication was done for THP-1 and S-Hela cells.

## Cell confluency and morphology imaging

Cell population growth in medias with increasing amounts of D$_2$O was measured by imaging cell confluency on a plate over ~3 days. Confluency was monitored using the IncuCyte live cell analysis imaging system by Sartorius. Imaging was carried out every 3 hr using 4× objective and confluency was analyzed using IncuCyte S3 2017 software's standard settings. The confluency data were then used to calculate confluency doubling time, which was used as a proxy for cell cycle duration. For validations of cell morphology in mitosis, cells were imaged every 2 min with 20× objective using the IncuCyte. Metaphase-anaphase transition was detected by simultaneous imaging of the FUCCI cell cycle sensor on FITC channel. The example images were overlaid and processed in the IncuCyte S3 2017 software.

## Chemical perturbations, cell cycle synchronizations, cell cycle analyses

Chemical treatment concentrations are indicated in main text and supplier details can be found in the key resources table. For SMR experiments, the cells were not pretreated with the chemical inhibitors, and all treatments started at the beginning of each SMR experiment. The treatment time until mitosis therefore varies between experiments, with typical range being 2–4 hr. Apart from EIPA treatments, all SMR experiments with chemical inhibitors lasted only for one cell division.

For cell cycle synchronization in G2 and mitosis, cells growing at ~400.000 cells/ml confluency were first arrested in G2 using 5 µM RO-3306 treatment. For L1210 and BaF3 cells, this treatment lasted for 6 hr; for THP-1 cells this treatment lasted for 20 hr. The cells were then washed twice with cold PBS and moved into fresh media containing either 5 µM RO-3306 (for G2 arrest) or 5 µM STLC (for prometaphase arrest). For L1210 and BaF3 cells, this treatment lasted for 1 hr; for THP-1 cells this

treatment lasted for 2 hr. After this, the cells were either fixed for cell cycle analysis or immunolabeled with anti-LAMP-1 antibody in drug containing media. Cell cycle arrest in G2 and mitosis was validated using flow cytometry-based detection of DNA and p-histone H3 (Ser10) labeling, as detailed before (*Miettinen et al., 2019*). Results in *Figures 4A and 5C* and *Figure 5—figure supplement 2A&C* represent the fraction of cells that were p-histone H3 (Ser10) positive and contained 4N DNA content, as evaluated based on a comparison to freely proliferating cells.

## Measurements of cell surface localized LAMP-1

In L1210 and BaF3 cells, the cell surface localized LAMP-1 protein was detected using immunolabeling with rat monoclonal anti-LAMP-1 antibody conjugated to Alexa Fluor 488 (Thermo Fisher Scientific; RRID:AB_657536). The cells were moved to fresh media (containing chemical treatments, where applicable) and incubated with the antibody at 1:50 dilution for 40 min on ice, after which the cells were washed twice with fresh media and either imaged or measured using flow cytometry. For antibody isotype control, we used rat monoclonal IgG2a kappa Isotype Control antibody conjugated to Alexa Fluor 488 (Thermo Fisher Scientific; RRID:AB_493963). For THP-1 cells, the immunolabeling protocol was otherwise identical, but the immunolabeling was carried out with mouse monoclonal anti-LAMP-1 antibody conjugated to Alexa Fluor 488 (Thermo Fisher Scientific; RRID:AB_2016657) or with corresponding isotype control (RRID:AB_470230).

When imaging the LAMP-1 immunolabeled cells, the last 20 min of antibody labeling was carried out together with DNA labeling using NUCLEAR-ID red DNA stain at 1:250 dilution (Enzo Life Sciences, Cat#ENZ-52406). For microscopy, we used DeltaVision wide-field deconvolution microscope with standard filters (FITC, APC), 100× objective and an immersion oil with a refractive index of 1.516. For each image, we collected 15 z-slices with 0.2 µm spacing, deconvolved the images using SoftWoRx 7.0.0 software and selected a middle z-slice for visualization. For flow cytometry-based detection of LAMP-1, DNA was not stained and LAMP-1 immunolabeling was detected using BD Biosciences flow cytometer LSR II HTS with excitation laser at 488 nm and emission filter at 530/30.

Image quantifications were carried out manually from single z-layers by cropping out each cell and calculating the average LAMP-1 labeling intensity per cell in ImageJ (version 2.0.0-rc-69/1.52 p). z-layers with intracellular labeling were rare and not used for analysis. Cells were classified as interphase, early, or late mitotic cells based on their DNA morphology. After this, the background signal was removed from the cell signals, and the LAMP-1 labeling of each cell was normalized to the average intensity observed in all interphase cells within the same image (typical image contained ~19 interphase cells).

## SMR operation for dry mass and dry mass density measurements

Basic SMR build and operation was identical to that reported before (*Kang et al., 2020*; *Kang et al., 2019*; *Miettinen et al., 2019*). The concept of dry mass and dry mass density measurements is detailed in *Feijó Delgado et al., 2013*. For dry mass and dry mass density measurements, the bypass channel on one side of the SMR cantilever is filled with $H_2O$-based media and the other side with $D_2O$-based media. For end-point single-cell measurements, the cells are first loaded on the $H_2O$-based media side of the device at ~300.000 cells/ml concentration. A cell is then flown through the SMR to obtain a measurement in the $H_2O$-based media. On the other side of the cantilever the cell is diluted in $D_2O$-based media, which is in excess to the $H_2O$-based media to maximize dilution. In a typical end-point measurement, the cell spends 5–8 s in the $D_2O$-based media. The cell is then flown back through the SMR to obtain a second measurement, this time in the $D_2O$-based media. After this, both sides of the cantilever are flushed with fresh media and cell(s) and the cycle is repeated. For single-cell monitoring, the protocol is the same as for end-point measurements, except that the cell is maintained for ~55 s in the $H_2O$-based media and ~15 s in $D_2O$-based media. Time in $D_2O$-based media is kept longer than in end-point measurements, as this increases the long-term stability of single-cell trapping. Importantly, the SMR baseline frequency is recorded prior and after every cell measurement. The SMR baseline frequency is directly proportional to media density, which allows us to adjust each measurement for local mixing of $H_2O$ and $D_2O$-based medias around the cell.

## Calibration and calculation of dry mass and dry mass density from SMR data

The two buoyant mass measurement (SMR frequency response to a cell flowing through the cantilever) and media density measurements (SMR frequency baseline immediately before and after the cell flows through the cantilever) were used to calculate dry mass, dry volume, and dry mass density according to *Equation 1*, assuming that extracellular fluid (culture media) density is equal to the density of water ($H_2O$ or $D_2O$) inside the cell (*Feijó Delgado et al., 2013*). In order to calibrate the SMR baseline, sodium chloride solutions of known density were measured at RT and used to generate a calibration curve for SMR baseline. In addition, the buoyant mass measurements were calibrated using 10 µm diameter monodisperse polystyrene beads (Thermo Fisher Scientific, Duke Standards), where the volume and density of the measured particle are known (*Mu et al., 2020*).

## Dry mass and dry mass density measurement resolution estimations

The precision of dry mass and dry mass density measurements was evaluated by repeatedly measuring the same, non-growing particle. As there are many sources of noise, three types of non-growing particles were used: a fixed L1210 cell measured at +37°C (fixation in 4% PFA in PBS for 10 min followed by a wash with PBS), a live L1210 cell measured at +4°C, and a 10 µm diameter polystyrene bead measured at +37°C. Each particle was tracked for dry mass and dry mass density similarly to that described above for live cells. For each experiment, 1 hr long section of the trace was used to calculate the measurement precision, presented as the coefficient of variability. When evaluating experiment-specific measurement noise (*Figure 3—figure supplement 2D*), we fitted the last 1 hr of dry mass data in interphase with a linear fit and calculated the SD of the residuals. Note that this calculation includes both technical and biological noise.

## Intracellular water volume approximation using the SMR

Approximate intracellular water volume of FUCCI L1210 cells was measured in the SMR by growing the cells in culture media containing 40% OptiPrep Density Gradient Medium (Sigma-Aldrich). The OptiPrep containing media has a solution density which is closer to cell's dry mass density than the density of water, thus making the SMR-based buoyant mass measurement predominantly sensitive toward intracellular water volume (*Equation 1*; *Son et al., 2015*). Apart from the different media, the SMR was operated as before (*Miettinen et al., 2019*), and the measurements were calibrated using 10 µm diameter monodisperse polystyrene beads (Thermo Fisher Scientific, Duke Standards).

## Data analysis, smoothing, and plotting

For analyses of mitotic dry mass and dry mass density dynamics, each cell trace analysis was limited to data starting from 1 hr before mitotic entry and ending at 2 hr after mitotic entry or at cell division. The very last datapoint prior to abscission was excluded in order to avoid analysis of cells during the final abscission. The traces were then normalized to their average value observed in the 1 hr period before mitotic entry, and the traces were overlayed. When plotting the dry mass, dry volume, or dry mass density mean ± SD values for each control and/or treatment condition, the moving and locally weighted average was plotted together with identical length moving standard deviation. For quantifications of dry mass, dry volume, or dry mass density changes in early mitosis, we identified the time point in mitosis where, on average, each control/treatment reached the minimum dry mass. The average of five datapoints at the maximal effect size was then compared the average of five datapoints immediately prior to mitotic entry (inclusive of the indexed mitotic entry timepoint). The difference between these two values is defined as 'early mitotic change'. Note that the maximal effect size was always achieved prior to metaphase-anaphase transition. Same smoothing and quantification approach was also used for processing the intracellular water volume measurements. For single-cell examples, the cut-out traces were smoothened using LOESS smoothing.

The mode of cell growth was analyzed by comparing an exponential and a linear fit to the raw dry mass data in interphases. AIC was used to define which fit is better (*Liu et al., 2020*). Dry mass growth rates were analyzed by fitting a 1 hr long section of data with a linear line, using the slope of the line as the growth rate and using the average dry mass within the 1 hr section as the dry mass corresponding to the growth rate. This fit was moved across the interphase of each cell to generate cell growth traces, which were plotted as a function of dry mass and averaged together to display population

average growth behavior. Only cells with full cell cycle (from birth to division) were used for analyses of exponential growth. All data processing and smoothing was done using MATLAB (versions 2017b and 2020b), and figures were plotted using MATLAB and Origin (version 2021b).

## Statistics

The statistical tests used, the results of these tests and detailed information about replicate numbers are indicated in each figure and figure legend. All statistical tests were carried out using Origin (version 2021b) software. Appropriate sample sizes were not computed before experiments and sample sizes were defined by practical considerations (e.g. experimental time requirements). We did not blind-control experiments, as most work was carried out by a single individual. When trapping a cell in the SMR for dry mass and dry mass density measurements, we trap randomly the first normal-sized cell that flows through the SMR after cells are flushed into the system.

## Source data

All dry mass and dry mass density traces of live cells, including those where cells are treated with chemical inhibitors, are available in *Figure 3—source data 1*.

## Acknowledgements

We thank Dr Ye Zhang for assistance with data analysis and Prof. Marco Cosentino Lagomarsino for helpful feedback on the manuscript. SRM received funding and support from the MIT Center for Cancer Precision Medicine, Virginia, and DK Ludwig Fund for Cancer Research, Cancer Systems Biology Consortium (U54CA217377), and Cancer Center Support (core) Grant P30-CA14051 from the National Cancer Institute.

## Additional information

### Competing interests

Scott R Manalis: is a co-founder of Travera and Affinity Biosensors, which develop technologies relevant to the research presented in this work. The other authors declare that no competing interests exist.

### Funding

| Funder | Grant reference number | Author |
|---|---|---|
| National Cancer Institute | P30-CA14051 | Scott R Manalis |
| MIT Center for Cancer Precision Medicine | | Scott R Manalis |
| Virginia and D.K. Ludwig Fund for Cancer Research | | Scott R Manalis |
| National Cancer Institute | U54CA217377 | Scott R Manalis |

The funders had no role in study design, data collection and interpretation, or the decision to submit the work for publication.

### Author contributions

Teemu P Miettinen, Conceptualization, Data curation, Formal analysis, Investigation, Methodology, Supervision, Validation, Visualization, Writing – original draft, Writing – review and editing; Kevin S Ly, Data curation, Formal analysis, Visualization, Writing – review and editing; Alice Lam, Investigation, Writing – review and editing; Scott R Manalis, Funding acquisition, Project administration, Resources, Supervision, Writing – review and editing

### Author ORCIDs

Teemu P Miettinen http://orcid.org/0000-0002-5975-200X
Alice Lam http://orcid.org/0000-0003-4332-4761

Scott R Manalis [ID] http://orcid.org/0000-0001-5223-9433

**Decision letter and Author response**
Decision letter https://doi.org/10.7554/eLife.76664.sa1
Author response https://doi.org/10.7554/eLife.76664.sa2

## Additional files

### Supplementary files
• Transparent reporting form

### Data availability
Supporting source data files contain all key data (dry mass and dry mass density traces) used in this study.

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
