## [Editor Report]

The authors measure dry mass, dry volume and the density of the dry mass in growing and proliferating mammalian cells at high temporal resolution and with high precision. Using this method to study mitotic cells, the authors show that some cells lose dry mass early in mitosis by a mechanism involving exocytosis. This work improves upon the authors' method to measure the mass of single cells and its thought-provoking conclusion is that dividing cells 'clean out' their contents to give the daughter cells a fresh start.

---

## [Decision Letter]

**Decision letter after peer review:**

Thank you for submitting your article "Single-cell monitoring of dry mass and dry density reveals exocytosis of cellular dry contents in mitosis" for consideration by *eLife*. Your article has been reviewed by 3 peer reviewers, and the evaluation has been overseen by a Reviewing Editor and Jonathan Cooper as the Senior Editor. The following individual involved in review of your submission has agreed to reveal their identity: Matthieu Piel (Reviewer #1).

Essential revisions:

1) All the reviewers request that you clarify the definition of dry mass, dry density, dry mass density, and dry volume. Below you will see the comments of each reviewer on this point, which should help you to clarify these terms.

2) A mathematical model could help to clarify the interpretation of your results. Referee 3 suggests a simple mathematical model that would roughly estimate how the cell composition (e.g., the contents of lipids vs proteins) should change and what the composition of the lost (secreted) components should be to provide the observed changes to the dry mass and density, given the existing information on average cell composition and the densities of different biomolecules (lipids, sugars, proteins, etc).

3) Some of the drug treatments require more replicates to provide more conclusive answers. In particular, Figure 4E-F: Given the SD for barasertib and EIPA cells, the authors need more than ~6-10 cells to draw conclusions based on treatment effects. That barasertib treatment does not affect dry mass and dry density changes is not yet convincing.

Other comments:

1) The authors show that their measure does not depend on the time spent in D2O, which is important. But it means that the water needs only a few seconds to exchange. Is that expected given the permeation of water through the plasma membrane and the size of the cell? I did not make the calculation, but it seems a bit short. Can the authors give also the theoretical estimate based on the permeation of water through the membrane?

2) For the measure of the exponential growth rate, the authors could use the same approach as in Cadart et al. Nat Com 2018, which might be more robust than fitting on the single cell growth curve: measuring the growth speed (dm/dt) on short periods of time all along the cell growth trajectories and for all cells and plotting dm/dt versus m. This should show clearly that growth is exponential on average.

3) The authors show that the changes in dry mass and dry volume are very different frorm cell to cell. How does that relate to the extent of mitotic swelling, which is happening at the same time? Is there a correlation between the two: do cells that swell more also lose more dry mass and/or dry volume (or increase more their dry density)? Plotting one against the other might help understanding whether the two processes are coupled. The authors show that when cells are treated with EIPA, they do not swell but still show an increase in dry density. But it does not mean that the two processes are not coupled in control cells. EIPA treatment is changing the capacity of the cell to swell (for unclear reasons – what it does the most directly is to change the intracellular pH) and it could decouple the two processes but it does not mean that might not share a common driving mechanism (in addition to mitotic entry of course). So it might be an informative plot to make.

4) At the bottom of page 10 the authors discuss inhibiting 'cell elongation', which is a characteristic of anaphase cells. But they also then inhibit the next step, called furrowing, which is an even more drastic change of shape (when the cell divides in two). They could also mention it as a change of shape that does not affect the dry mass density.

5) For the readers who are interested in more details of these experiments, it might be interesting to give, in the sup figures, the distribution of volumes and dry mass of the cells which have been treated with the various drugs (for example used in Figure 4), as they are given only normalized in the main figure. It would be interesting to know whether the drugs are changing these values (for example just before cells enter mitosis, do they enter mitosis with the cell volume/dry mass/density when they have been treated with EIPA before). This is more curiosity from this referee…so the authors might as well choose not to show all that.

6) In the discussion, line 322, the authors mention high temporal resolution without phototoxicity as an advantage of their method over quantitative phase microscopy. QPM also allows high temporal resolution without phototoxicity. But the main point I think is that it does not measure the same thing! In particular, it does not give access to this 'dry volume' parameter.

7) The results seem to be cell type dependent and it is quite surprising that the dry mass loss is different for L1210 expressing the FUCCI marker compared with wild-type L1210. Can the authors speculate on why this could be the case (e.g. different levels of exocytosis activity?). Quantification of other cell types in the main figure would be helpful. In line with this, the reduction in dry mass loss with Brefeldin A or Vacuolin-1 inhibition in Figure 6 is convincing but comparable to that observed in wild type cells, while the FUCCI cells show more dramatic decrease in cell mass. The authors should acknowledge this clearly in the text and soften the claim that chemical inhibition of lysosomal exocytosis prevents mitotic dry composition changes.

8) 'cell elongation' (page 10): do cells elongate in suspension?

9) Does the mitotic exocytosis described in this relate to previous observations that cells 'bleb' to help correct for asymmetric positioning of the mitotic spindle (see https://doi.org/10.1038/nature14496 for example?)

10) The method seems to affect the dynamics of mitosis. Line 149 page 6 states 'while limited D2O exposure appears to influence mitosis duration, it does not radically …..' This caveat should be mentioned again in the discussion.

---

## [Author Response]

Essential revisions:1) All the reviewers request that you clarify the definition of dry mass, dry density, dry mass density, and dry volume. Below you will see the comments of each reviewer on this point, which should help you to clarify these terms.

Thank you for this feedback. We apologize for the confusion caused by our terminology, which was based on the terminology used in our original manuscript (Feijo Delgado et al., 2013). We have now changed our terminology to be consistent with more recent works by others. We now define ‘dry density’ as dry mass divided by total volume, and we define our measurements of dry mass divided by dry volume as ‘dry mass density’. This change has been applied throughout our manuscript, including our manuscript title. In addition, we have added clearer definitions of each term to our Introduction and Measurement Method sections. Furthermore, we have minimized the use of the term ‘dry composition’ throughout our manuscript, as we now realize this may cause confusion to some readers.

More specifically, our introduction (page 3) now states: “Here, we introduce a new approach for monitoring single cell’s dry mass (i.e. total mass – water mass), dry volume (i.e. total volume – water volume), and density of the dry mass (i.e. dry mass / dry volume), which we will refer to as dry mass density.” These definitions are also repeated in our Measurement Method section (page 4), as many readers may look for the definitions in that section.

2) A mathematical model could help to clarify the interpretation of your results. Referee 3 suggests a simple mathematical model that would roughly estimate how the cell composition (e.g., the contents of lipids vs proteins) should change and what the composition of the lost (secreted) components should be to provide the observed changes to the dry mass and density, given the existing information on average cell composition and the densities of different biomolecules (lipids, sugars, proteins, etc).

Thank you for this feedback. We fully agree that such calculations could be very useful in interpreting our results. We have now estimated how much lipids a theoretical cell would have to lose in order to explain the dry mass loss and dry mass density increase. This is explained in a new paragraph (Discussion section, page 13) as follows:

“The magnitude of dry mass density increase in mitosis was large. We have previously observed similar magnitude changes in dry mass density when perturbing proliferation in mammalian cell (Feijo Delgado et al., 2013). To provide some rough estimates of what kind of compositional changes would be required to achieve the dry mass loss and dry mass density increase, we carried out a back-of-the-envelope calculations. Assuming a typical mammalian cell composition and typical macromolecule dry mass densities (Alberts, 2008; Feijo Delgado et al., 2013), we calculated the degree of lipid loss needed to increase dry mass density by 2.5%. This suggested that cells would have to secrete ~1/3 of their lipid content in early mitosis. This could be achieved via lysosomal exocytosis of lipids. Lipid droplets, the main lipid storages inside cells, are frequently trafficked into and degraded in lysosomes (Singh et al., 2009), and lipid droplets can also be secreted via lysosomal exocytosis (Minami et al., 2022). However, it seems likely that the mitotic dry mass density increase also involves secretion of other low dry mass density components (e.g. lipoproteins, specific metabolites) and/or a minor, transient increase in high dry mass density components (e.g. RNAs, specific proteins) in early mitosis. Indeed, CDK1 activity has been suggested to drive a transient increase in protein and RNA content in early mitosis (Asfaha et al., 2022; Clemm von Hohenberg et al., 2022; Miettinen et al., 2019; Shuda et al., 2015).”

3) Some of the drug treatments require more replicates to provide more conclusive answers. In particular, Figure 4E-F: Given the SD for barasertib and EIPA cells, the authors need more than ~6-10 cells to draw conclusions based on treatment effects. That barasertib treatment does not affect dry mass and dry density changes is not yet convincing.

Yes, we agree that the Barasertib treatment experiments would require more replicates in order to have full confidence that the drug does not influence dry mass and dry mass density behavior in mitosis. After consulting the *eLife* editors, we have instead softened our conclusions to acknowledge that Barasertib treated cells could still have changes in their dry mass behavior when compared to control cells. Our updated Results section (page 9)

“The mitotic changes in cell’s dry mass and dry mass density cannot be fully explained by cell elongation and furrowing, although cell elongation and furrowing (or Aurora B activity) could still have an influence on dry mass and dry mass density dynamics in mitosis.”

We have also changed our figure 4 title to “Mitotic dry mass loss and dry mass density increase do not require morphological changes”, and we have removed the sentence from our abstract that claimed that dry mass changes are independent from cytokinesis.

Other comments:1) The authors show that their measure does not depend on the time spent in D2O, which is important. But it means that the water needs only a few seconds to exchange. Is that expected given the permeation of water through the plasma membrane and the size of the cell? I did not make the calculation, but it seems a bit short. Can the authors give also the theoretical estimate based on the permeation of water through the membrane?

Thank you for the feedback, this is a great suggestion. We have now carried out rough estimations of the water exchange time, as based on the water permeability of aquaporin channels, the typical abundance of aquaporin channels and the typical volume of a L1210 cell. We have also added more references that provide direct experimental support for the time required to exchange cellular water content. We detail these at the end of our Measurement method section (page 5) as follows:

“Our measurement principle involves the cells exchanging their intracellular water content between H2O and D2O, and this exchange has to be complete for our measurements to reflect the dry mass and dry mass density of cells. Direct experimental evidence suggests that cells normally exchange their water content within ~1 second (Potma et al., 2001; Quirk et al., 2003; Zhao et al., 2008). To support this, we carried out back-of-the-envelope calculations on the intracellular water exchange times, as based on reported aquaporin water permeability (Gravelle et al., 2013; Hashido et al., 2007; Yang and Verkman, 1997), abundance (Denker et al., 1988) and a cell water volume of ~1000 fl (approximate water volume for a L1210 cell). This suggested that cells are capable of exchanging all of their water volume within 0.1 – 2.5 s, depending on the reported aquaporin water permeability. Notably, the true intracellular water exchange time may differ from this due to intracellular water diffusion, water molecule interactions with other biomolecules and additional means of water transport in and out of the cell (additional aquaporin isoforms, endo- and exocytosis, etc.).”

2) For the measure of the exponential growth rate, the authors could use the same approach as in Cadart et al. Nat Com 2018, which might be more robust than fitting on the single cell growth curve: measuring the growth speed (dm/dt) on short periods of time all along the cell growth trajectories and for all cells and plotting dm/dt versus m. This should show clearly that growth is exponential on average.

Thank you for the suggestion, we have now added the recommended plot, which clearly shows that growth rates (pg/h) increase as cell mass increases within L1210 cell interphases. This is detailed on pages 6 and 7 of our manuscript, stating that: “Consistent with exponential growth, instantaneous dry mass growth rates (pg/h) increased as a function of dry mass (Figure 2E).”

3) The authors show that the changes in dry mass and dry volume are very different frorm cell to cell. How does that relate to the extent of mitotic swelling, which is happening at the same time? Is there a correlation between the two: do cells that swell more also lose more dry mass and/or dry volume (or increase more their dry density)? Plotting one against the other might help understanding whether the two processes are coupled. The authors show that when cells are treated with EIPA, they do not swell but still show an increase in dry density. But it does not mean that the two processes are not coupled in control cells. EIPA treatment is changing the capacity of the cell to swell (for unclear reasons – what it does the most directly is to change the intracellular pH) and it could decouple the two processes but it does not mean that might not share a common driving mechanism (in addition to mitotic entry of course). So it might be an informative plot to make.

Thank you for the feedback. Unfortunately, our measurement capabilities do not allow us to simultaneously measure the dry mass (and dry mass density) and the total volume of a cell, and developing such methods would require major engineering work. Thus, we cannot at the moment correlate the degree of mitotic cell swelling to the degree of mitotic dry mass loss. That being said, we agree with the reviewer’s comment, and we have now added the following conclusion to our manuscript (Results section, page 10):

“The mitotic dry mass and dry mass density changes do not require mitotic cell swelling. However, it remains possible that these two phenotypes share a common driving mechanism.”

4) At the bottom of page 10 the authors discuss inhibiting 'cell elongation', which is a characteristic of anaphase cells. But they also then inhibit the next step, called furrowing, which is an even more drastic change of shape (when the cell divides in two). They could also mention it as a change of shape that does not affect the dry mass density.

Thank you for pointing this out, as we do indeed prevent furrowing using Aurora B inhibition. We have now pointed this out to the reader in the relevant Results section (page 9).

5) For the readers who are interested in more details of these experiments, it might be interesting to give, in the sup figures, the distribution of volumes and dry mass of the cells which have been treated with the various drugs (for example used in Figure 4), as they are given only normalized in the main figure. It would be interesting to know whether the drugs are changing these values (for example just before cells enter mitosis, do they enter mitosis with the cell volume/dry mass/density when they have been treated with EIPA before). This is more curiosity from this referee…so the authors might as well choose not to show all that.

Thank you for the suggestion, but we have chosen not to add such figures. Cell dry mass and dry volume vary significantly between individual cells even when examining all cells specifically at mitotic entry, especially since the cells are measured over a timespan of many months. In the absence of significantly larger datasets, we find such examinations to have little value. The normalization we utilize in the figures allows us to examine the mitosis specific effects despite this inherent size variability. However, all our raw data is attached to our manuscript, so these analyses can be carried out by others who consider them more important.

6) In the discussion, line 322, the authors mention high temporal resolution without phototoxicity as an advantage of their method over quantitative phase microscopy. QPM also allows high temporal resolution without phototoxicity. But the main point I think is that it does not measure the same thing! In particular, it does not give access to this 'dry volume' parameter.

Thank you for pointing this out. We have indeed acknowledged in our discussion that our method can measure the dry volume of cells, which makes our method unique. We have also adjusted our text in the discussion to avoid claims that QPM could not achieve high temporal resolution.

7) The results seem to be cell type dependent and it is quite surprising that the dry mass loss is different for L1210 expressing the FUCCI marker compared with wild-type L1210. Can the authors speculate on why this could be the case (e.g. different levels of exocytosis activity?). Quantification of other cell types in the main figure would be helpful. In line with this, the reduction in dry mass loss with Brefeldin A or Vacuolin-1 inhibition in Figure 6 is convincing but comparable to that observed in wild type cells, while the FUCCI cells show more dramatic decrease in cell mass. The authors should acknowledge this clearly in the text and soften the claim that chemical inhibition of lysosomal exocytosis prevents mitotic dry composition changes.

Thank you for this feedback. The cell type-dependency of our results is indeed quite surprising, although we have previously also observed that mitotic energy metabolism can differ significantly even between the wt and FUCCI expressing L1210 cells (Kang, et al., 2020, Nature Comms). For all of these phenotypes, we have not found any solid evidence that would explain the cell type-dependency. It is possible that baseline levels of endo- and exocytosis could explain the difference in dry mass loss in mitosis, but in the absence of more evidence, we would prefer to avoid this discussion in our manuscript. However, we completely agree with the reviewer that our conclusions should be rephrased. Our updated Results section (page 11) now states that:

“Inhibitors of lysosomal exocytosis decrease mitotic dry mass loss and dry mass density increase” (section title), and that “Lysosomal exocytosis is, at least partly, responsible for the loss of dry mass and increased dry mass density in early mitosis. However, other mechanisms, including other forms of exocytosis, may also be involved, because inhibition of lysosomal exocytosis did not prevent all dry mass loss in mitosis, and because our results are limited to a single cell line.”

In addition, we have now stated in our discussion (pages 13) that:

“Other mechanisms, especially other forms of exocytosis, may also be involved, as we were not able to completely block the dry mass loss in early mitosis by preventing lysosomal exocytosis.”

8) 'cell elongation' (page 10): do cells elongate in suspension?

Yes, even in suspension cells the phenotype observed during early anaphase corresponds to cell elongation. This is evident, at least to some degree, in our Figure 4B, where we display timelapse imaging of our suspension grown L1210 FUCCI cells (imaging is carried out in suspension culture). Yet, to avoid confusion, we now call this “cell elongation and furrowing”, as the detailed reviewer comments noted that cell furrowing is a distinct cytokinetic event that we also inhibit.

9) Does the mitotic exocytosis described in this relate to previous observations that cells 'bleb' to help correct for asymmetric positioning of the mitotic spindle (see https://doi.org/10.1038/nature14496 for example?)

This is an interesting hypothesis, but the timing of cell dry mass loss seems to precede the cell blebbing. More specifically, our data shows that cell mass loss takes place predominantly from mitotic entry (prophase) to metaphase-anaphase transition, whereas the polar relaxation of actin cortex and the consequent cell blebbing takes place predominantly in anaphase. Notably, we have examined polar relaxation in our L1210 FUCCI cells (Kang, et al., Nature Methods, 2019) and the timing of this phenotype is the same in our L1210 cells as in the other model systems examined in the literature (e.g. Rodrigues, et al., Nature, 2015). We therefore consider it unlikely that the cell blebbing typically observed in mitosis would be part of the mechanism causing dry mass loss. Yet, the mitotic exocytosis provides excess plasma membrane (as we note in our discussion), and this excess membrane could enable or even promote blebbing

10) The method seems to affect the dynamics of mitosis. Line 149 page 6 states 'while limited D2O exposure appears to influence mitosis duration, it does not radically …..' This caveat should be mentioned again in the discussion.

Thank you for pointing this out. In our updated Discussion section (on page 12), we now state that:

“However, our measurement approach also decreased the duration of mitosis and cytokinesis, which could potentially influence the dry mass dynamics we observe in mitosis.”